# Multiclass Segmentation of Concrete Surface Damages Using U-Net and DeepLabV3+

Patrick Nicholas Hadinata [1], Djoni Simanta [1], Liyanto Eddy [1,*] and Kohei Nagai [2]

1 Department of Civil Engineering, Universitas Katolik Parahyangan, Bandung 40141, Indonesia
2 Institute of Industrial Science, The University of Tokyo, Tokyo 153-8505, Japan
* Correspondence: liyanto.eddy@unpar.ac.id

**Abstract:** Monitoring damage in concrete structures is crucial for maintaining the health of structural systems. The implementation of computer vision has been the key for providing accurate and quantitative monitoring. Recent development uses the robustness of deep-learning-aided computer vision, especially the convolutional neural network model. The convolutional neural network is not only accurate but also flexible in various scenarios. The convolutional neural network has been constructed to classify image in terms of individual pixel, namely pixel-level detection, which is especially useful in detecting and classifying damage in fine-grained detail. Moreover, in the real-world scenario, the scenes are mostly very complex with varying foreign objects other than concrete. Therefore, this study will focus on implementing a pixel-level convolutional neural network for concrete surface damage detection with complicated surrounding image settings. Since there are multiple types of damage on concrete surfaces, the convolutional neural network model will be trained to detect three types of damages, namely cracks, spallings, and voids. The training architecture will adopt U-Net and DeepLabV3+. Both models are compared using the evaluation metrics and the predicted results. The dataset used for the neural network training is self-built and contains multiple concrete damages and complex foregrounds on every image. To deal with overfitting, the dataset is augmented, and the models are regularized using L1 and Spatial dropout. U-Net slightly outperforms DeepLabV3+ with U-Net scores 0.7199 and 0.5993 on F1 and mIoU, respectively, while DeepLabV3+ scores 0.6478 and 0.5174 on F1 and mIoU, respectively. Given the complexity of the dataset and extensive image labeling, the neural network models achieved satisfactory results.

**Keywords:** convolutional neural network; damage detection; semantic segmentation; deep learning; computer vision

## 1. Introduction

Periodic and reliable inspections of structural properties in infrastructure systems are required to maintain their service capabilities and to ensure public safety. Engineers have constructed a methodology for conducting structural assessment, which is known as structural health monitoring (SHM). The stages in SHM are detecting the existence of damage, locating the damage, identifying the types of damage, and quantifying the severity of damage. Conventionally, the aforementioned stages rely on subjective human assessment, which greatly depends on the inspector's experiences. Due to the uncertainty, such procedures have traditionally been considered unsustainable due to frequent inaccurate damage assessment, which ultimately leads to inefficient budget and materials allocation. To tackle this issue, more quantitative and advanced statistical approaches have been proposed over the decades. Although there are many types of structural damages, automatic detection methods using computer vision techniques for concrete surface cracks were developed extensively. Methods such as thresholding, which segment the cracks based on the intensity of the color and continuous nature of the object, have been proposed [1–3]. Some related advancements, such as using edge detection and texture descriptor, were also proposed

mboxciteB4-applsci-2192849,B5-applsci-2192849,B6-applsci-2192849. The results were fast and accurate for localized and controlled conditions. However, to make the model extensively deployable, it is crucial to consider real-world data scenarios. Such data often come with significant complexity in terms of foreground objects, which the traditional thresholding and edge detection method has difficulty dealing with. Moreover, it is also difficult to expand the model to deal with more types of damages [7]. Therefore, more-robust methods were developed using deep learning methods. One of the popular deep learning methods in computer vision technologies due to its accuracy and flexibility is the convolutional neural network (CNN).

The CNN is an improvement of the traditional artificial neural networks by introducing sliding window techniques of a kernel that can efficiently extract features without using too many parameters. The main purpose of a CNN is to learn and classify features from images through supervised training. A CNN typically adopts a contracting shape that effectively reduces feature dimension and highlights important features. Earlier layers mostly highlight coarse features, such as texture detection, while deeper layers highlight class-specific features [8]. The CNN has been adapted to some object recognition tasks, namely image classification, object detection, and image segmentation. Image classification simply classifies the entire image, object detection predicts the location of an object on an image using bounding boxes, and image segmentation classifies every pixel in an image to its corresponding class. One advantage of image segmentation over object detection is that it allows for fine-grained, pixel-level detection of objects. In the damage detection task, it can be very useful for the area detection of a damage, which is useful for measuring the damage properties, such as the length and width of surface cracks. Advanced morphological feature extraction methods to measure crack dimensions have been implemented using data from segmentation output [9,10].

A CNN built specifically for image segmentation outputs images with the same resolution as the input. In the typical CNN structure, the end layers consist of fully connected layers. A CNN for semantic segmentation replaces the fully connected layers with convolutional layers. As such, every layer is essentially convolutional. This method was introduced by Long et al. [11], resulting in an architecture called the fully convolutional network (FCN). This development was quickly adopted and modified to segment more complex objects. Inspired from the foundation design of the FCN, U-Net [12] was introduced with a more defined encoder–decoder structure, with multiple skip connections on every convolution level. The decoder in U-Net, especially, has more channels and layers than that in FCN. It was argued that, by passing information on every level from the encoder to the decoder, localization details will be recovered [12]. Since then, U-Net has been widely used to segment objects across many disciplines [13–15]. Inspired by the encoder–decoder method, DeepLabV3 [16], which was previously a naïve decoder architecture, was upgraded to DeepLabV3+ [17] with an encoder–decoder architecture. Depthwise separable convolution was introduced, which ultimately reduces computation costs on a large architecture [17]. It achieved the highest mIoU on Pascal VOC 2012 dataset [17]. In conclusion, U-Net and DeepLabV3+ have been widely used in various segmentation tasks due to their flexibility.

In the application for damage detection in infrastructure, there are several research areas that focus on different structures and different types of structural changes. In the area of pavement damage detection, methods such as thresholding using the Otsu method [18], deep learning segmentation [19], and classification of damage types using a generative adversarial network [20] have been developed. In addition to methods for detecting simple visible damage such as cracks, some methods to measure the strain of structures were also developed by Tang et al. [21]. This research will focus on damage detection in concrete structures. In the area of concrete structures, a block-level classification of concrete surface cracks was implemented by Cha et al. [22]. To provide more detailed classification, a pixel-level method was adopted using an FCN for the binary segmentation of surface cracks [23–25]. Due to the common problem in FCNs, which is smoothened edges and coarser granularity of cracks [26], U-Net and DeepLabV3+ were implemented to tackle the

problem, and both U-Net and DeepLabV3+ performed better than FCN [26] for binary crack segmentation. Some modifications were also proposed to the state-of-the-art architectures for crack detection. Huyan et al. [27] modified U-Net such that it requires fewer parameters to run for binary crack segmentation. Zhou et al. [28] implemented U-Net with a multiscale feature fusion module and Resnet block in the encoder. However, there are other types of damage on concrete surfaces such as spalling and voids. The issue with the previous research [26] is that the CNN often considers voids as cracks. Therefore, the CNN has been adapted to ignore voids that occurred on concrete surfaces [29].

Instead of just ignoring the damages other than cracks, a CNN can be adapted to differentiate the damages into different classes. Li et al. [7] have successfully implemented FCN for detecting multiple damages on concrete structures, namely efflorescence, cracks, voids, and spallings. However, their dataset mostly consists of a single type of damage in an image due to the difficulty of collecting a dataset that contains multiple damages in an image. Deng et al. [30] and Rubio et al. [31] constructed a dataset that mostly consists of multiple damages in an image. Their dataset consists of two concrete surface damages, namely rebar exposure and delamination. Their results contributed significantly toward understanding the behavior of CNNs starting from dataset construction to model evaluation. However, most of the dataset does not deal with a highly complex foreground. To realize the implementation of such models in a complex environment, it is important to understand the boundaries of such CNN models in a highly complex foreground, especially with relatively very limited data. A previous study [26] only dealt with simple binary crack images using a publicly available dataset constructed by Yang et al. [23]. Therefore, the broad objective of this research is to apply encoder–decoder CNN models based upon a newly constructed complex dataset for multiclass damage segmentation. Some feature engineering and regularization techniques will also be recommended to deal with the dataset.

Since this study uses a newly designed dataset, a classic encoder–decoder architecture will be used to investigate the training behavior. Therefore, the first objective of this research is to investigate the applicability of U-Net and DeepLabV3+ to detect multiple damages on concrete surfaces. Both architectures will be reconstructed and compared. Since the limitation of related research [7] was difficulty in constructing a dataset with multiple types of damage in an individual image, a custom dataset mostly consisting of multiple damages in every image will be constructed. The research conducted by Deng et al. [30] already deals with multiple damage in an image and their damages are rebar exposure and delamination. Therefore, this paper will deal with different types of damages, namely cracks, voids, and spallings. Moreover, the dataset is also purposely built to deal with a complex foreground, which corresponds to a highly complex real-world scenario.

A previous study [26] showed that the Dice loss function successfully dealt with the class imbalance problem. In this study, because of the class imbalance issue, a multiclass generalized Dice loss (GDL) function is used. Therefore, the second objective of this research is to investigate the use of the GDL function in multiclass damage segmentation. Finally, the third objective of this study is to investigate the use of two regularization techniques, namely L1 regularization and spatial dropout [32] on multiclass damage segmentation for complex images. For evaluation purposes, every model with different loss functions and different regularization combinations will be qualitatively and quantitatively compared.

The rest of this paper is organized as follows. Section 2 will discuss the dataset's construction, CNN architectures, applied regularizations, evaluation metrics, and training algorithms. Section 3 will discuss the results of the CNN training, which comprises the quantitative comparisons, qualitative comparisons, and limitations of the work. Finally, Section 4 concludes the paper.

## 2. Materials and Methods

This section will discuss the methods for data construction and the training schematic for the neural networks. Overall, the proposed methodology is depicted in Figure 1.

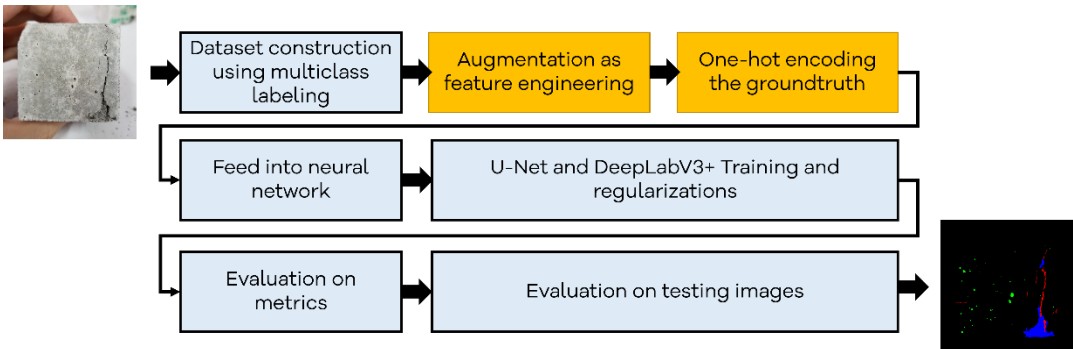

**Figure 1.** Proposed methodology.

## 2.1. Datasets

The dataset comprised 100 images with a 1024 × 1024-pixel resolution. All images were taken by the authors using a Samsung Galaxy Note 8 or Fujifilm X100. Note that other camera sensors can be used to extend the dataset diversity. In this study, there are four classes in the dataset, namely background, crack, spalling, and void, which are examples of commonly occurring damages on concrete surfaces. Background, crack, spalling, and void in every image were manually annotated using Adobe Photoshop and labeled as 0, 1, 2, and 3, respectively. To deal with the common overfitting problem in neural network training, the dataset was further expanded using the data augmentation method. Before being augmented, the dataset was shuffled and portioned as 80% for training and 20% for validation. In the augmentation process, all 100 images were rotated three times at 90°. Every rotation produced another 80 images for the training dataset and 20 images for the validation dataset. After being rotated three times, the total training dataset comprised 320 images, while the total validation dataset held 80 images. The authors decided to augment the images only by using a 90° angle. It is important to augment the dataset with high variability, because rotating the images by less than 90 degrees would only make a more similar image, which corresponds to a highly correlated dataset [33]. The data augmentation technique is illustrated in Figure 2.

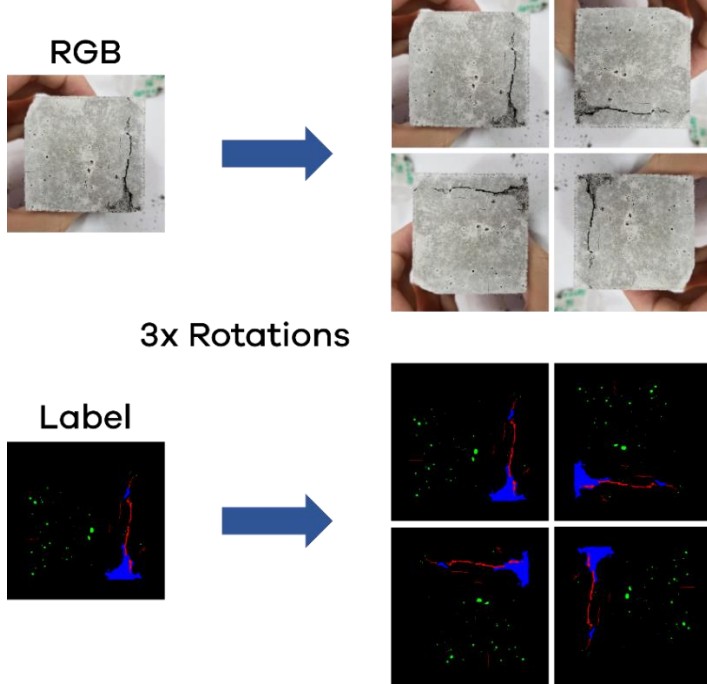

**Figure 2.** Data augmentation using rotation.

The dataset was also configured to deal with complex background scenarios, for example, hands, building accessories, non-focused objects, stones, leaves, and many more, which are very commonly occurring. The surface damages were taken from cylindrical concrete specimens, mortar specimens, beam test specimens, and concrete walls. Some examples from the dataset are depicted in Figure 3.

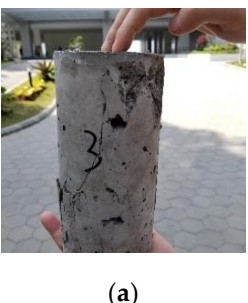 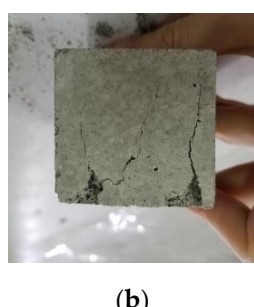 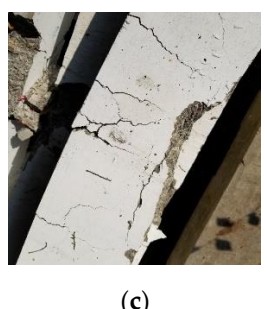 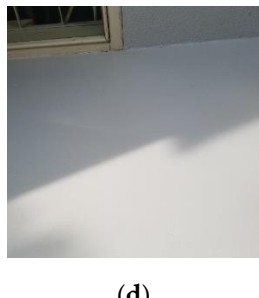

(**a**)  (**b**)  (**c**)  (**d**)

**Figure 3.** (**a**) Dataset from cylindrical concrete specimen; (**b**) dataset from mortar specimen; (**c**) dataset from beam specimen; (**d**) dataset from concrete wall.

Before being fed into the neural network, the resolution of the dataset was first resized to 512 × 512 resolution, because the original 1024 × 1024 resolution consumed too much GPU memory. Then, the image was normalized by dividing the pixel value by 127.5 and subtracting it by 1. Therefore, the pixel value of every image was between 1 and −1. The annotation was divided into 4 slices because there were 4 classes as depicted in Figure 4. This is necessary to deal with the SoftMax activation function in the network output. For decent visualization, the class of each damage was represented with vibrant colors, while the background was represented by black. Cracks were visualized in red, spallings were visualized in blue, and voids were visualized in green. The visualization is illustrated in Figure 5.

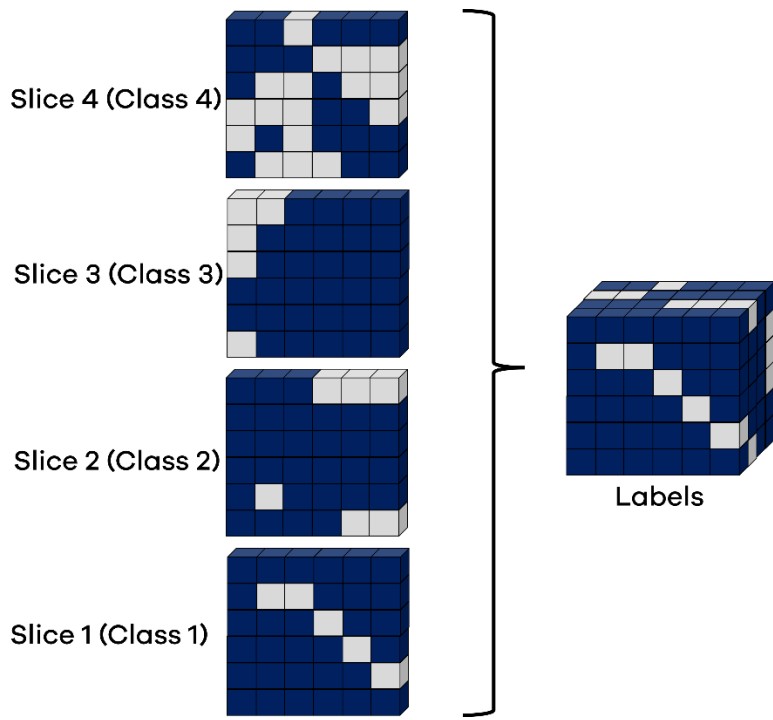

**Figure 4.** Class slices. White pixel is the labeled class, and blue pixel is the other classes.

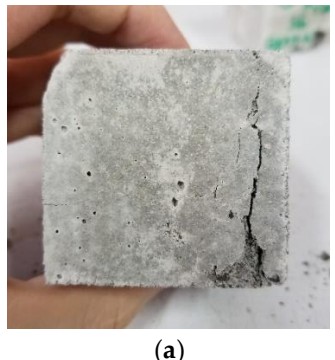
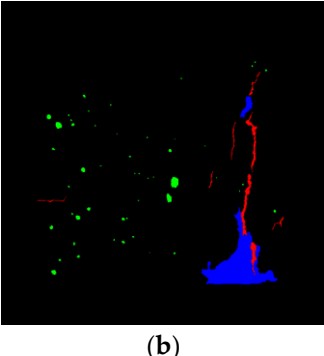

(**a**)                                                                            (**b**)

**Figure 5.** (**a**) RGB Image; (**b**) ground truth image. Voids are visualized as green, spallings are visualized as blue, and cracks are visualized as red.

*2.2. Model Architecture*

2.2.1. U-Net

The main feature of U-Net is its U-shaped architecture and concatenation on every convolution level. The encoder section of U-Net is very similar to that of VGG-16 architecture. The encoder part of U-Net uses 2 blocks of convolution, followed by max-pooling operations. The feature channels start from 64, which then double after every max-pooling until they reach 1024 channels. The decoder section is symmetrical with the encoder section with 2 blocks of convolution, followed by transpose convolution operations. Transpose convolution, typically known as deconvolution, is simply a reversed type of convolution. While bilinear upsampling has a fixed kernel to upsample a feature map, transpose convolution's kernel is flexible and can be trained to fill in data loss during the upsampling process. One advantage of transpose convolution is the process can be non-linear, which can effectively eliminate unnecessary features. Transpose convolution is illustrated in Figure 6. The encoder is connected with the decoder using the concatenate feature. After two convolution operations in the encoder, the feature map was transferred to the decoder, just after each transpose convolution. The illustration for U-Net is shown in Figure 7.

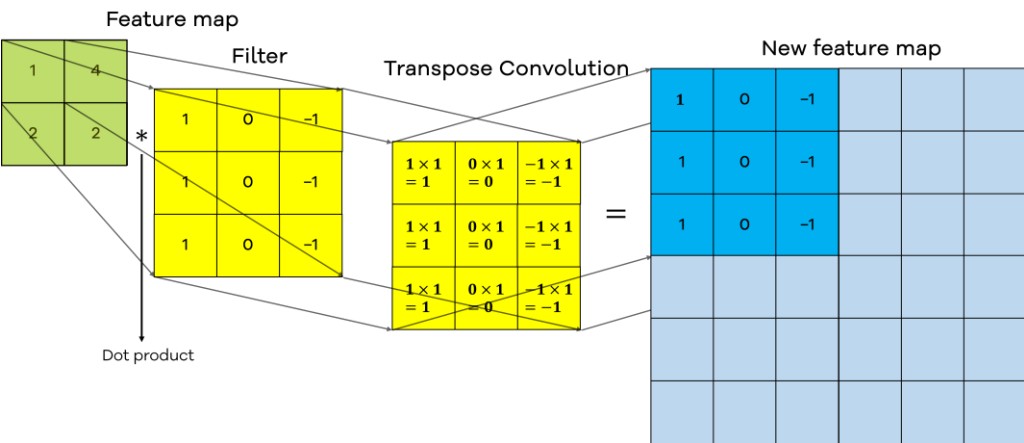

**Figure 6.** Transpose convolution.

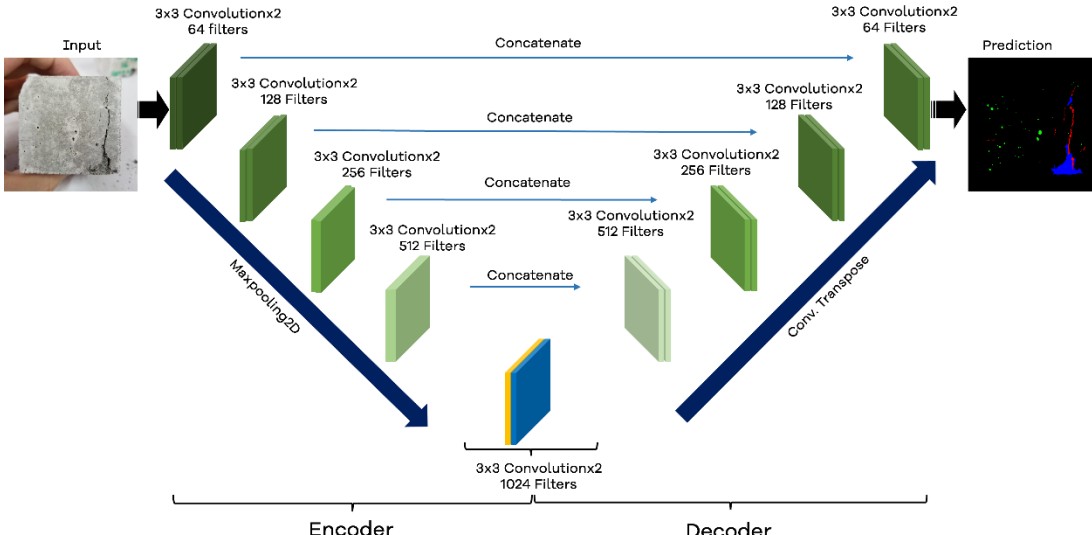

**Figure 7.** U-Net architecture.

### 2.2.2. DeepLabV3+

The main feature of DeepLabV3+ is the modified, aligned Xception backbone and the ASPP module. Chen et al. [17] modified the Xception [34] by adding more layers, extra batch normalization, and ReLu after every depthwise separable convolution operation. The modified Xception structure is illustrated in Figure 8. Depthwise separable convolution consists of a depthwise convolution and a pointwise convolution [34]. The depthwise convolution performs the convolution operation using a different filter on every channel. While the pointwise convolution simply performs a single-pixel dot product as illustrated in Figure 9. This reduces the computation cost compared to standard convolution and is very effective in a large network such as the DeepLabV3+. Depthwise separable convolution is illustrated in Figure 8.

The ASPP module uses depthwise atrous separable convolution of different rates, pointwise convolution with 256 feature channels, and average pooling. All layers from the ASPP module were concatenated and passed to the decoder. In the decoder, the feature map was further concatenated with information from the earlier layer of the encoder. Finally, a simple two-times convolution and bilinear upsampling were implemented. Illustration for DeepLabV3+ is depicted in Figure 10.

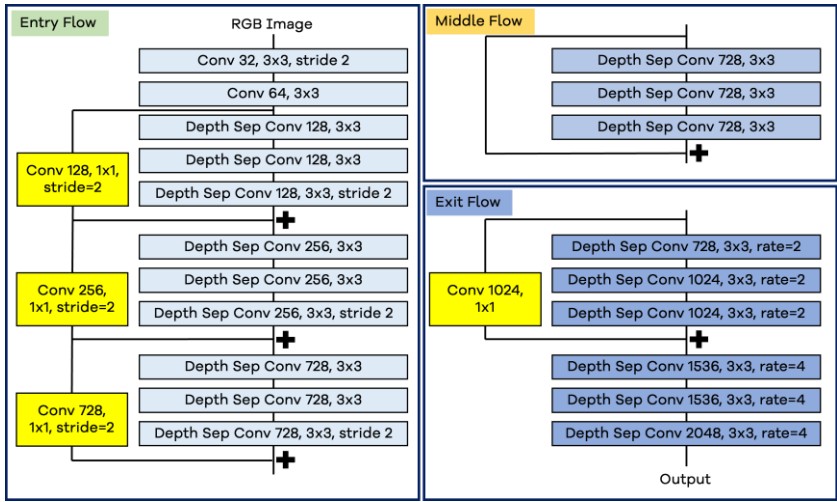

**Figure 8.** Modified Xception backbone.

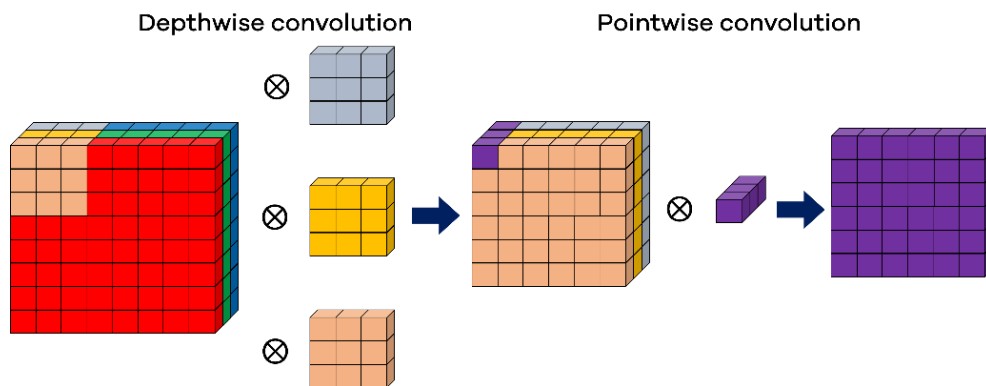

**Figure 9.** Depthwise separable convolution.

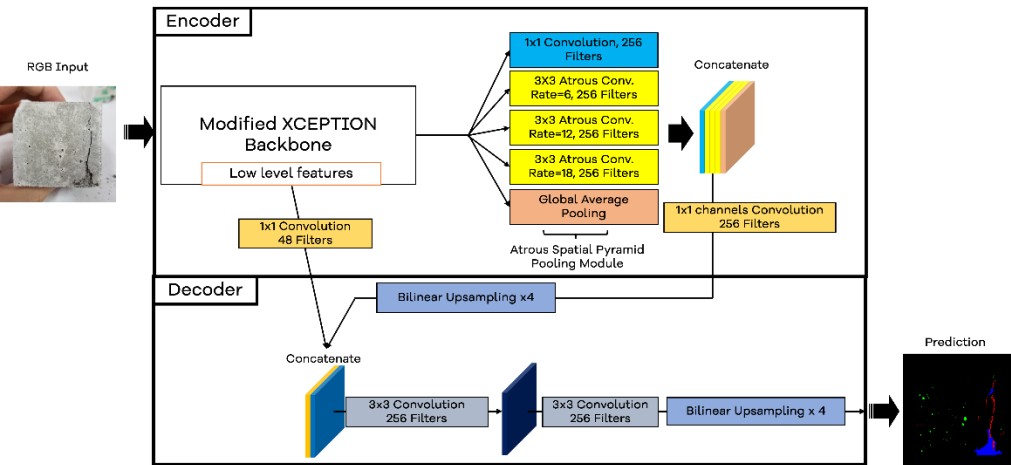

**Figure 10.** DeepLabV3+ architecture.

*2.3. Regularizations*

There are two points to highlight regarding the dataset. First, the size of the dataset is relatively small. Second, the images in the dataset consist of a complex foreground. These circumstances could lead to an overfitting in the training and validation. Therefore, it is common to use regularization techniques to ensure good neural network generalization. There were two regularization techniques used in this study, namely, spatial dropout and L1 regularization.

2.3.1. Spatial Dropout

Dropout [35] was introduced to combat overfitting due to random deactivation of a feature, which effectively acts as model averaging. However, in the image recognition task, neighboring pixels are often highly correlated. Therefore, regular dropout may not be effective in regularizing the model. As such, spatial dropout was introduced by Tompson et al. [32]. Spatial dropout deactivates the entire feature map instead of individual pixels. It was demonstrated that replacing regular dropout with spatial dropout improves performance in CNN models [36]. In both U-Net and DeepLabV3+, the spatial dropout rate was set at 0.2 and is implemented after every convolution layer, inspired by [37]. The difference between spatial dropout and regular dropout is illustrated in Figure 11.

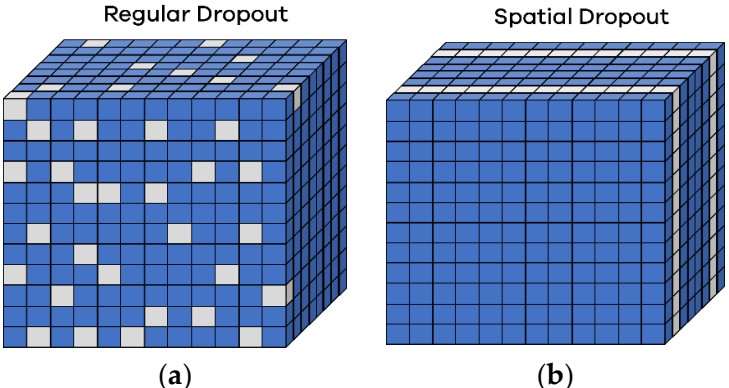

**Figure 11.** (**a**) Regular dropout channel deactivation; (**b**) spatial dropout channel deactivation.

### 2.3.2. L1 Regularization

L1 regularization, also known as least absolute shrinkage and selection operator (LASSO) regression, regularizes the neural network by adding a weight penalty. The L1 regularization in this paper was implemented only to the kernel, where most parameters are inside the high-dimensional feature vector. L1 regularization for the Dice loss function is described in Equation (1). The lambda value needs to be chosen carefully. Too large a value of lambda can result in underfitting, while too small a value might not regularize the model. The lambda value for both U-Net and DeepLabV3+ was set at $1 \times 10^{-5}$.

$$\text{GDL}_{\text{L1}} = \text{GDL} + \lambda |w| \tag{1}$$

where GDL is the generalized Dice loss, $w$ is the weight of a kernel, and $\lambda$ is a constant value ranging between 0 and 1, set experimentally.

### 2.4. Evaluation Metrics

For the classification task, there are four commonly used evaluation indexes, namely true positive (TP), true negative (TN), false positive (FP), and false negative (FN). In the multiclass segmentation problem, the evaluation index for each class is calculated individually, meaning that, when a specific class is evaluated, it considers other classes as the foreground. For example, when the parameters are calculated for cracks, TP means that the truth is crack and the prediction is also crack, TN means that the truth is non-crack and the prediction is also non-crack, FP means that the truth is non-crack while the prediction is crack, and FN means that the truth is crack while the prediction is non-crack.

In this segmentation task, there is class imbalance due to the domination of the background area. Therefore, evaluation metrics used for measuring the models' performance cannot favor the dominant class. In this research, accuracy could not be used as the main metric, instead Dice coefficient and intersection over union (IoU) were used for the comparison purpose. Dice coefficient and IoU are defined in Equations (2) and (3), respectively.

$$\text{Dice coefficient} = \frac{1}{S} \sum_{s=1}^{S} \sum_{k=0}^{K} 2 \frac{p_s^k g_s^k + \epsilon}{p_s^k + g_s^k + \epsilon} \tag{2}$$

$$\text{IoU} = \frac{1}{S} \sum_{s=1}^{S} \sum_{k=0}^{K} \frac{p_s^k g_s^k + \epsilon}{p_s^k + g_s^k - (p_s^k g_s^k) + \epsilon} \tag{3}$$

where $S$ is the total number of samples, $K$ is the total number of classes, $p_s^k$ is the prediction on a specific sample $s$ and class $k$, $g_s^k$ is the corresponding grountruth, $w_k$ is the class rebalancing weight that needs to be set manually, and $\epsilon$ is a small constant to prevent zero division.

### 2.5. Adam Optimizer

As the name suggests, the optimizer's main function is to optimize the neural network model toward a relatively minimum error by computing the necessary learning steps in a gradient descent algorithm. These learning steps adjust the weight and bias in the neural network using the backpropagation method. Adaptive moment estimation (Adam) [38] was adapted by taking the advantages of RMSProp [39] and SGD with momentum [40]. Adam has been successfully implemented for most deep learning tasks, and it has been assessed as one of the best-performing optimizers [41–43]. The Adam update rule is defined in Equations (4)–(9).

$$g_j = -\nabla L \big| w_{i,j} \tag{4}$$

$$s_j^i = \rho_1 s_j^{i-1} + (1 - \rho_1) g_j \tag{5}$$

$$r_j^i = \rho_2 r_j^{i-1} + (1 - \rho_2)(g_j)^2 \tag{6}$$

$$\hat{s}_j^i = \frac{s_j^i}{1 + \rho_1} \tag{7}$$

$$\hat{r}_j^i = \frac{r_j^i}{1 + \rho_2} \tag{8}$$

$$W_j^{i+1} = W_j^i + \frac{\epsilon \hat{s}_j}{\delta + \sqrt{\hat{r}_j}} \tag{9}$$

where $g_j$ is the gradient of a specific learning step; $s_j^i$ is the first-order moment estimate, $r_j^i$ is the second-order moment estimate; $\hat{s}_j^i$ is the first-order bias correction; $\hat{r}_j^i$ is the second-order bias correction; $W_j^i$ is weight from the kernel; $\rho_1$ is a hyperparameter with a default value of 0.9; $\rho_2$ is a hyperparmeter with a default value of 0.999; $\epsilon$ is the constant learning rate, which needs to be adjusted accordingly; and $\delta$ is a constant to prevent zero division with a default value of $10^{-7}$.

### 2.6. Generalized Dice Loss

The GDL is specialized for the imbalanced multiclass dataset. Sudre et al. [44] implemented it as a loss function from the generalized Dice score. It can be configured to focus on specific classes instead of the background. The GDL is defined in Equation (10).

$$\text{GDL} = 1 - \frac{1}{S} \sum_{s=1}^{S} \sum_{k=0}^{K} 2 \frac{p_s^k g_s^k w_k + \epsilon}{(p_s^k + g_s^k) w_k + \epsilon} \tag{10}$$

where $w_k$ is the class rebalancing weight that needs to be set manually. In this research, due to the class imbalances, $w_k$ for background is set as 0.1 and that for damage is set as 1.

### 2.7. Hardware and Software Configurations

Deep learning training, especially for image recognition tasks, takes advantage of the GPU core. GPU processing greatly reduces computation time by passing images as batches. With the addition of tensor cores, the capability of batch training is also increased. The Gradient Paperspace machine learning platform was chosen as a cloud computing platform due to its affordable computation options. The GPU used in this study was NVIDIA QUADRO RTX 5000 with 3072 Cuda Cores, 384 Tensor Cores, and 16 GB of GDDR6 VRAM. The gradient machine learning notebook was equipped with 30 GB of RAM and 15 GB of storage, which is sufficient for this deep learning task.

Due to its simplicity and efficient computation, TensorFlow was selected as the software API. Both U-Net and DeepLabV3+ were reconstructed by the authors using the TensorFlow functional API.

## 3. Results and Discussions

The training process for each network took approximately one hour. The training process for DeepLabV3+ was slightly longer than that of U-Net due to its larger parameters. After training, the class-specific IoU, class-specific F1, mean IoU, and mean F1 were calculated. In each testing result, the IoU for each class was displayed.

### 3.1. U-Net and DeepLabV3+ Training Results

### 3.1.1. U-Net Training Results

The F1 curve and IoU curve for training and validation of U-Net are shown in Figure 12. U-Net converged fast in the beginning of the training and then plateaued at around 63% to 70% after about 20 epochs in terms of the F1 score. Based on the training and validation curve, it can be concluded that U-Net generalizes very well on the validation data.

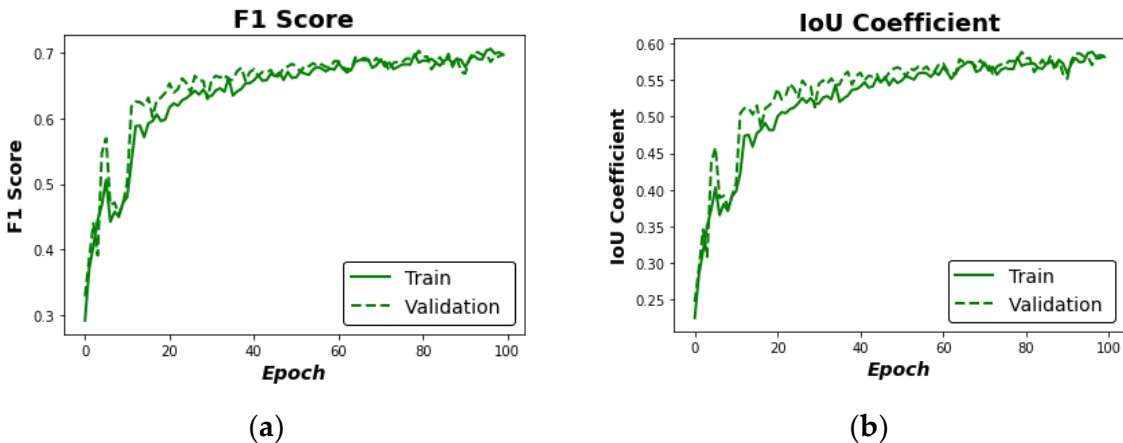

**Figure 12.** (**a**) U-Net F1 training curve; (**b**) U-Net IoU training curve.

### 3.1.2. DeepLabV3+ Training Results

DeepLabV3+ curves for training and validation are shown in Figure 13. Based on the close proximity of the training and validation curves, it can be concluded that DeepLabV3+ generalizes really well on unseen data.

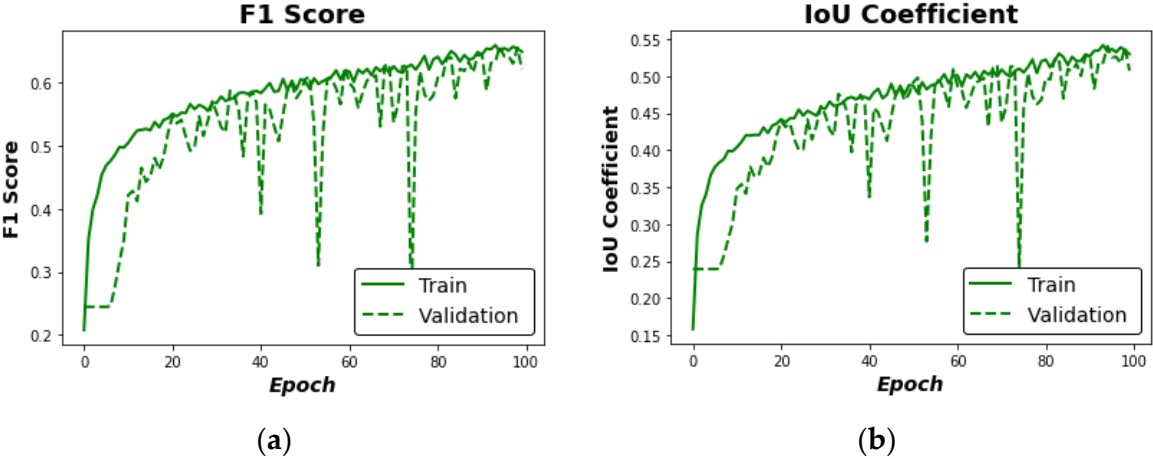

**Figure 13.** (**a**) DeepLabV3+ F1 training curve; (**b**) DeepLabV3+ IoU training curve.

### 3.1.3. Quantitative Comparison

Table 1 shows the final measurement of every evaluation metric. The F1 and mIoU for U-Net were 0.7199 and 0.5993 respectively, while the F1 and mIoU for DeepLabV3+ were 0.6478 and 0.5174 respectively. Based on these results, U-Net clearly outperformed DeepLabV3+. The only difference between U-Net and DeepLabV3+ is that U-Net tends to have better detection rate for the spalling class than for the crack class, while DeepLabV3+ tends to have better detection rate for the crack class than for the spalling class. The void class had the lowest values of F1 and mIoU for both models. This is arguably due to the large number of false negatives, caused by extensive labeling of small voids in the training data. Both models typically cannot detect voids that are smaller than four pixels in diameter. These results also suggest that the background class needs less attention than the other classes. Therefore, weighting 0.1 for background to the loss function is sufficient.

**Table 1.** U-Net and DeepLabV3+ training results comparison.

| Metrics | U-Net | DeepLabV3+ |
|---|---|---|
| Accuracy | 0.9800 | 0.9744 |
| Mean IoU | 0.5993 | 0.5174 |
| Mean F1 | 0.7199 | 0.6478 |
| Background IoU | 0.9828 | 0.9711 |
| Crack IoU | 0.4787 | 0.3979 |
| Spalling IoU | 0.6303 | 0.3769 |
| Void IoU | 0.3052 | 0.3237 |
| Background F1 | 0.9913 | 0.9854 |
| Crack F1 | 0.6475 | 0.5693 |
| Spalling F1 | 0.7732 | 0.5474 |
| Void F1 | 0.4677 | 0.4891 |

### 3.1.4. Qualitative Comparison

Figure 14 shows the prediction results of U-Net and DeepLabV3+ alongside the corresponding ground truth. Based on the result, U-Net and DeepLabV3+ successfully ignored the complex background and still had decent performance for damage detection. Based on the prediction results, U-Net can segment smaller objects and has a more accurate detection of boundaries. DeepLabV3+ often ignores small objects and overall has more-smoothened edges than U-Net. This result is also reflected on the quantitative comparison where the values of mean IoU and mean F1 of DeepLabV3+ are lower than those of U-Net. This phenomenon can be seen closely in Figure 15, where in the regions highlighted in yellow, U-Net detects small cracks and voids while DeepLabV3+ does not.

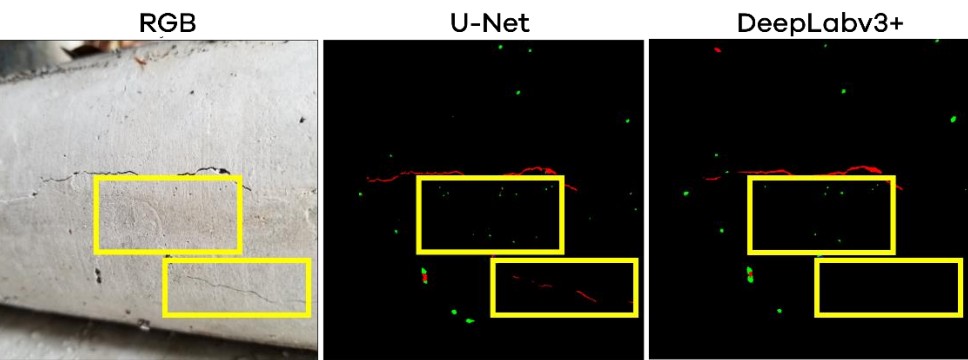

**Figure 14.** Models' prediction on validation data.

**Figure 15.** U-Net and DeepLabV3+ comparison on fine-grained object segmentation.

### 3.2. Effect of Generalized Dice Loss

3.2.1. Quantitative Comparison

Table 2 shows the comparison of models when trained using categorical cross entropy loss or GDL. The mIoU and FI for U-Net trained using categorical cross entropy were only 0.4778 and 0.5819, respectively. The mIoU and F1 for DeepLabV3+ were also much lower when it was trained using categorical cross entropy. The mIoU and F1 for DeepLabV3+ were 0.3821 and 0.4716, respectively. This proves that GDL with the class-rebalancing weight properties performs better than the classic categorical cross entropy.

**Table 2.** U-Net and DeepLabV3+ trained using categorical cross entropy and generalized Dice loss.

| Model | Mean IoU | F1 |
|---|---|---|
| U-Net with categorical cross entropy loss | 0.4778 | 0.5819 |
| U-Net with generalized Dice loss | 0.5993 | 0.7199 |
| DeepLabV3+ with categorical cross entropy loss | 0.3821 | 0.4716 |
| DeepLabV3+ with generalized Dice loss | 0.5174 | 0.6478 |

3.2.2. Qualitative Comparison

Figure 16 shows the testing images for both categorical cross entropy and GDL. These images clearly show that GDL is better than categorical cross entropy in both U-Net and DeepLabV3+. Categorical cross entropy makes both U-Net and DeepLabV3+ ignore smaller classes. In this case, cracks are often ignored. This shows that the class weighting properties on GDL are useful on a highly imbalanced dataset. These results are also reflected on the quantitative comparison section where the mIoU and F1 for categorical cross entropy are significantly lower than those for GDL.

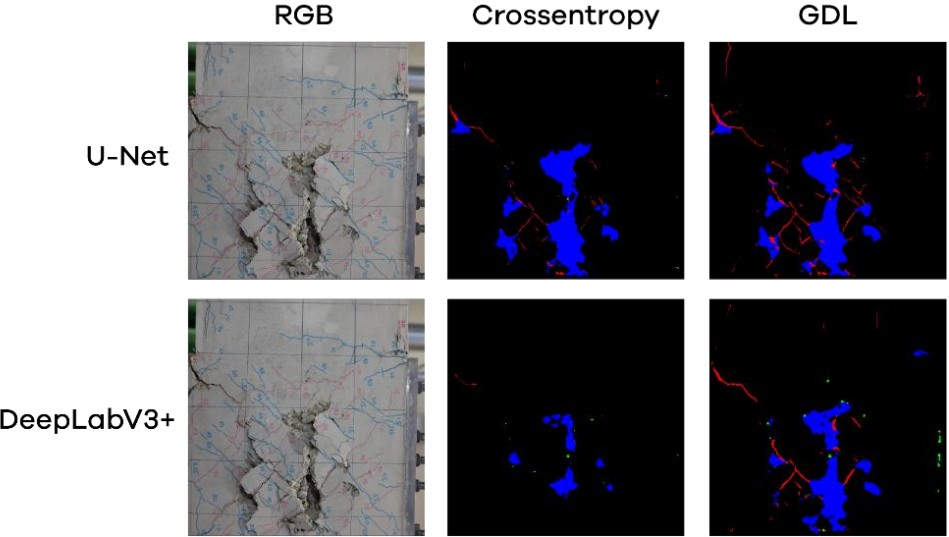

**Figure 16.** U-Net and DeepLabV3+ comparison on fine-grained object segmentation.

### 3.3. Effect of Spatial Dropout and L1 Regularization

3.3.1. Quantitative Comparison

Table 3 shows the training results of U-Net and DeepLabV3+, when several configurations of spatial dropout and L1 regularizations are implemented. Both U-Net and DeepLabV3+ trained without L1 and spatial dropout suffered from extreme overfitting, indicated by the big gap of the validation and training curve. Figure 17 shows the overfitting tendency of U-Net and DeepLabV3+ trained without L1 and spatial dropout. Both models started to overfit after epoch 10, indicated by the plateauing validation curve and the increasing training curve concurrently. The overfitting tendency of U-Net and DeepLabV3+ was slightly reduced when L1 regularization was applied as shown in Figure 18. Figure 19

shows that a similar case also happened when both models used spatial dropout. Both models slightly benefitted from the dropout techniques. Finally, when both L1 and spatial dropout were implemented in U-Net and DeepLabV3+, the models generalized very well on the validation data, which is shown by the overlapping of the validation and training curves in Figure 20. The evaluation metrics were also the highest when both L1 and spatial dropout were used, as shown in Table 3. Looking at the general line trends, some models had big fluctuations during training such as in Figure 17a around epoch 69. This might happen when an outlier ended up on a relatively small batch size. Since the dataset is relatively small and complex, the occurrence of outliers on a small, trained batch becomes larger. Note that the small batch size of four was chosen due to GPU memory limitation. However, the training quickly stabilized and returned to normal. Although the improvement on the validation data is not significant, the main contribution of these regularizations is stabilizing the discrepancy between training and validation performance. This is especially important when more data will be added in the future. Note that the dataset was divided into training and validation sets first before being augmented to make sure that the validation data did not generalize on the training data. Therefore, in a small and complex dataset with proper division between validation and training, the improvement after applying the regularization is significant. Since this is relatively complex data, which also corresponds to a possible higher fluctuation during training, adding a penalty to the weight might also reduce the chance of an exploding gradient.

**Table 3.** U-Net and DeepLabV3+ trained using no regularization, L1 regularization, spatial dropout, and L1 and spatial dropout.

| Model | Mean IoU | F1 Score |
|---|---|---|
| U-Net without L1 and spatial dropout | 0.5855 | 0.7065 |
| U-Net with L1 | 0.5940 | 0.7150 |
| U-Net with spatial dropout | 0.5801 | 0.7008 |
| U-Net with spatial dropout and L1 | 0.5993 | 0.7199 |
| DeepLabV3+ without L1 and spatial dropout | 0.5121 | 0.6194 |
| DeepLabV3+ with L1 | 0.5151 | 0.6284 |
| DeepLabV3+ with spatial dropout | 0.4851 | 0.5940 |
| DeepLabV3+ with spatial dropout and L1 | 0.5174 | 0.6478 |

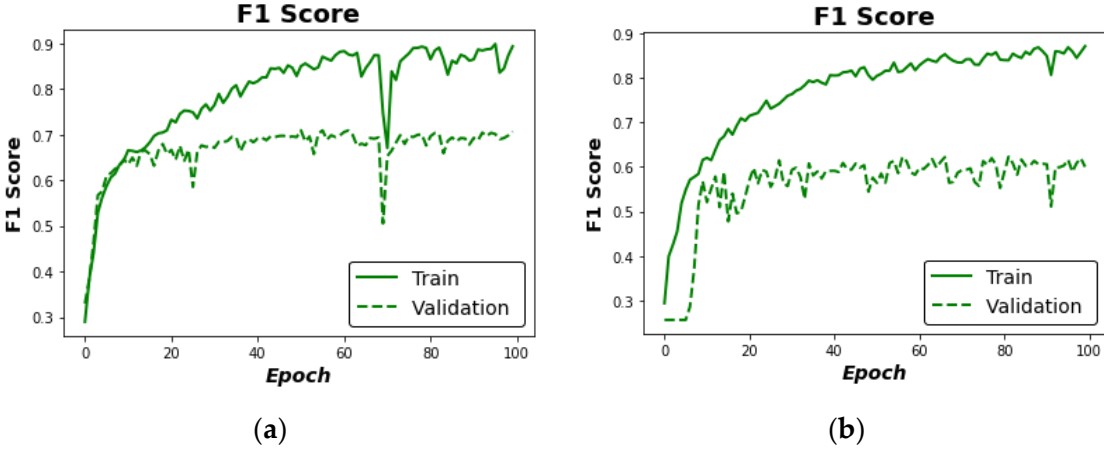

**Figure 17.** (**a**) U-Net without L1 and spatial dropout F1 curve; (**b**) DeepLabV3+ without L1 and spatial dropout F1 curve.

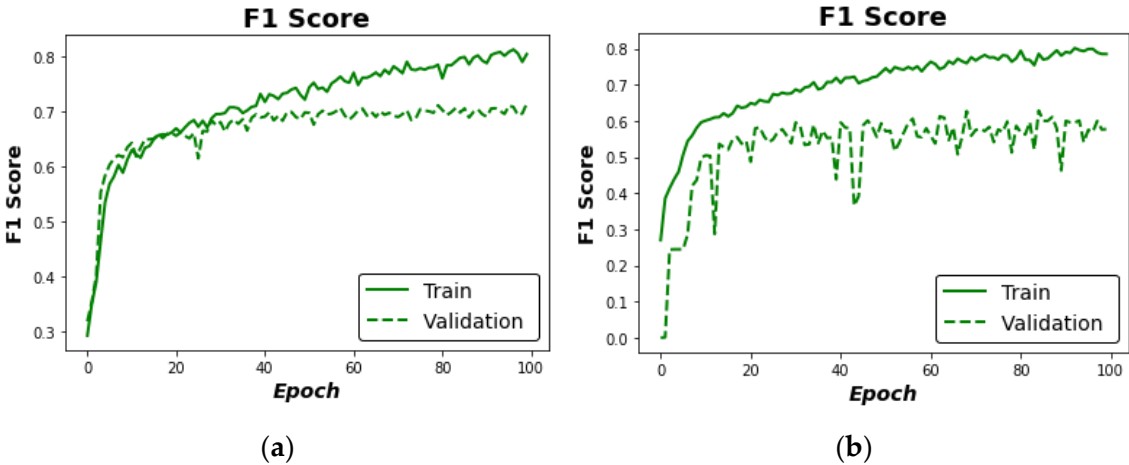

**Figure 18.** (**a**) U-Net with L1 F1 curve; (**b**) DeepLabV3+ with L1 F1 curve.

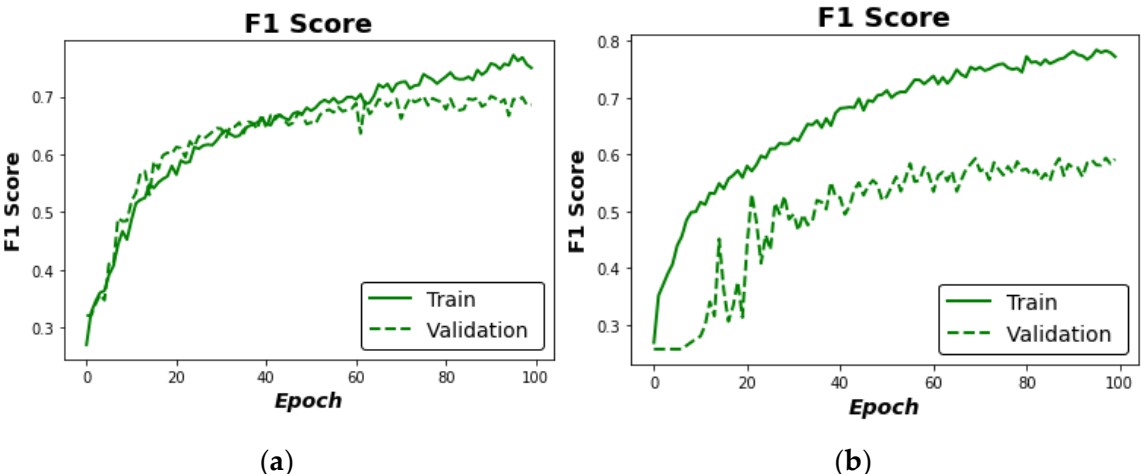

**Figure 19.** (**a**) U-Net with spatial dropout F1 curve; (**b**) DeepLabV3+ with spatial dropout F1 curve.

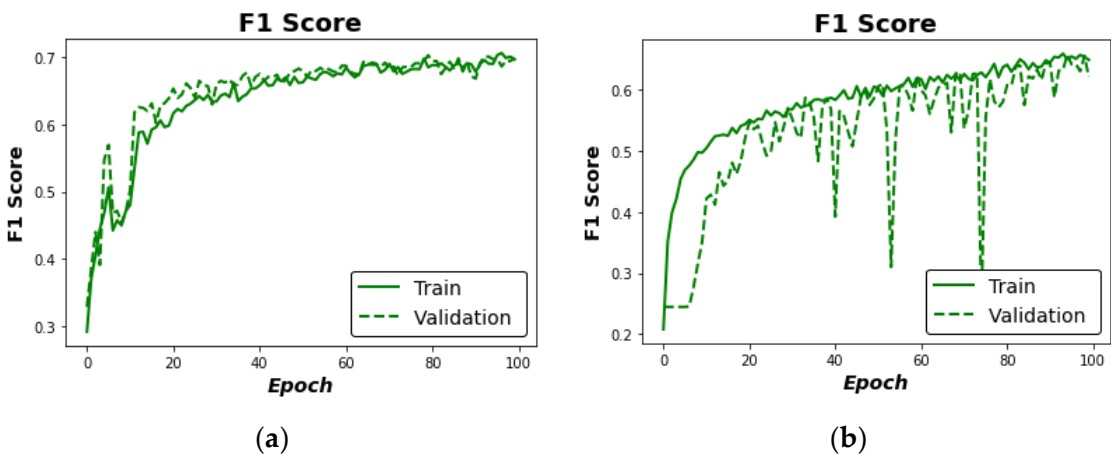

**Figure 20.** (**a**) U-Net with L1 and spatial dropout F1 curve; (**b**) DeepLabV3+ with L1 and spatial dropout F1 curve.

### 3.3.2. Qualitative Comparison

Figure 21 shows testing images based on the type of regularizations. When no regularization was used, both U-Net and DeepLabV3+ displayed an overfitting tendency. In U-Net, some black edges were often miscategorized as cracks, while in DeepLabV3+,

almost no spalling was detected. Both phenomena are shown in Figure 22a,b, respectively. When L1 regularization was used, both models performed better. Black edges were less miscategorized as cracks, and spalling was detected more correctly. When spatial dropout was used, almost the same results were obtained as when L1 regularization. Although, some cracks and spalling were less detected. Finally, when L1 and spatial dropout were used together, the models performed the best. However, some foreign objects are often miscategorized as cracks and voids. In this case, leaves were sometimes detected as cracks, and small rocks were sometimes detected as voids, as shown in Figure 22c.

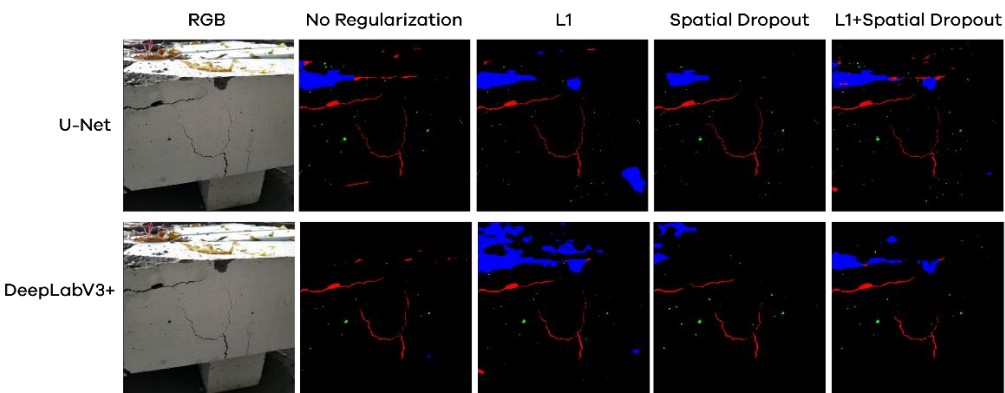

**Figure 21.** U-Net and DeepLabV3+ comparison on fine-grained object segmentation.

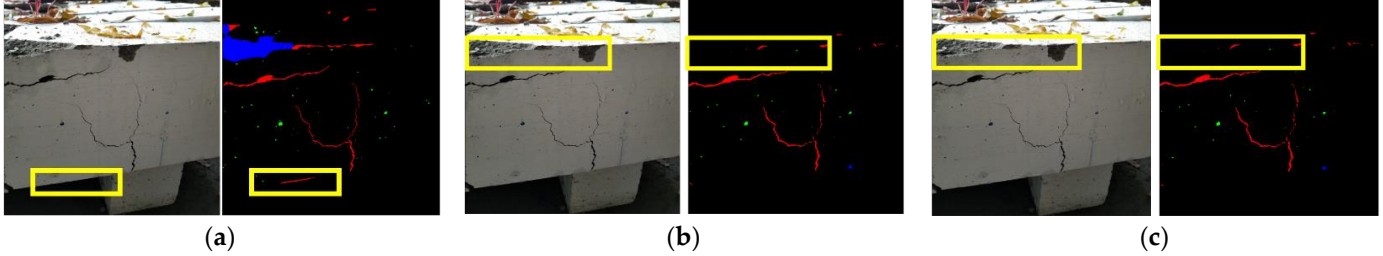

**Figure 22.** (**a**) Black edges detected as cracks on U-Net without regularization; (**b**) spalling not detected on DeepLabV3+ without regularization; (**c**) small rocks and leaves detected as damages on U-Net.

The qualitative results showed the same tendency as the quantitative results. From the quantitative comparison, the ranking was as follows: L1+spatial dropout > L1 > spatial dropout > no regularization. Based on the qualitative comparison, while the model with no regularization clearly performed the worst, the L1-only models were slightly better than the spatial-dropout-only ones. Hence, the qualitative ranking was identical to the quantitative one.

### 3.4. Study Limitations

This study uses a specific deep learning task called semantic image segmentation. The architecture used is two classical encoder–decoder CNN models, namely U-Net and DeepLabV3+. Since this is a new dataset with complex scenario, this study is focused not only on constructing a new architecture for the dataset but also on training the neural network in an end-to-end manner without any preprocessing since the focus is to study the behavior of the dataset in the CNN model. Therefore, no transfer learning is used in the backbone of U-Net and DeepLabV3+. Since the labeling of the dataset is relatively extensive, for example, very small cracks and voids are always labeled, the prediction often comes with false negatives. This affects the mIoU and F1 score of the final model. The miscategorization of some foreign objects such as leaves also affects the prediction metrics, because only a small number of datasets have leaves as foreign objects.

## 4. Conclusions

Based on the results and discussions, the following conclusions can be inferred:

1. U-Net and DeepLabV3+ were successfully reconstructed and trained for multiclass damage segmentation of concrete surfaces. Both models were implemented to differentiate background, crack, void, and spalling on concrete surfaces under complex scenarios.
2. U-Net generally performed better than DeepLabV3+. The F1 and mIoU of U-Net were 0.7199 and 0.5993, respectively, while the F1 and mIoU of DeepLabV3+ were 0.6478 and 0.5174, respectively. Overall, U-Net tended to have better fine-grained edge segmentation, while DeepLabV3+ tended to have coarser edge segmentation. All in all, both models performed well on a complex and limited dataset.
3. GDL successfully addressed the classes imbalance issue. By modifying the weight on the less important class, the F1 and IoU on the damage classes were improved. Compared to categorical cross entropy loss, which weighs every class the same, GDL was significantly better. Quantitatively, GDL scored around 10% better than categorical cross entropy in terms of F1 and mIoU, on both U-Net and DeepLabV3+. Qualitatively, categorical cross entropy often ignores classes with a smaller area, resulting in poor multiclass detection.
4. L1 and spatial dropout were used as the regularization techniques. Both techniques improved the overall model's convergence. Quantitatively, the use of either L1 or spatial dropout will improve the neural networks' convergence. However, by using both regularization techniques, both U-Net and DeepLabV3+ improved further. Qualitatively, when no regularization was used, U-Net often miscategorized black lines as cracks, and DeepLabV3+ often did not detect spalling. When only L1 or spatial dropout was used, both models improved their overall detection rate with the aforementioned misclassification seen less. When L1 and spatial dropout were used together, both models performed at their best.
5. This study provides a perspective to the deployment potential of damage inspection in infrastructure using deep learning. It is important to understand the behavior of such models in complex scenarios with limited data. For the future scope of studies, the dataset will be further expanded in terms of number of images and types of damages to ensure better generalization. A transfer-learning-based approach can also be implemented to significantly increase the model performance. For real-time detection, a new architecture and multisensor fusion can be constructed and trained using a faster GPU for inference.

**Author Contributions:** Data curation, P.N.H.; analysis, P.N.H.; writing—original draft preparation, P.N.H.; visualization, P.N.H.; methodology, P.N.H., D.S. and L.E.; supervision, D.S., L.E. and K.N.; writing—review and editing, D.S., L.E. and K.N. All authors have read and agreed to the published version of the manuscript.

**Funding:** This research received no external funding.

**Data Availability Statement:** The data presented in this study are openly available at "https://drive.google.com/file/d/1M76eI-TsgmfsKsMKTusONNcR_QEqEzdF/view?usp=sharing (accessed on 11 February 2023)" or "https://github.com/HadinataPN/Multiclass-Segmentation-of-Concrete-Surface-Damages-Using-U-Net-and-DeepLabV3- (accessed on 11 February 2023)".

**Conflicts of Interest:** The authors declare no conflict of interest.

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
