# Peer review of "Multiclass Segmentation of Concrete Surface Damages Using U-Net and DeepLabV3+"

_applsci, doi:10.3390/app13042398_

Round 1

Reviewer 1 Report

In this paper, the authors have created a concrete surface damage detection dataset and introduced two classical deep learning methods the U-Net and DeepLabV3+ for this task. The suggestions are listed as follows:

1.      The grammar and writings should be further improved. For example, page 1 line 12.

2.      Please describe the theoretical difference of the methods used in reference [7] and this article. Please note that CNN models have the ability to classify multiple types. Changing the binary dataset into multiple dataset cannot be viewed as the major theoretical contribution.

3.      Please describe the major contribution of this article over reference [7].

4.      The full name of GDL should be given where it firstly appears.

5.      More related works on concrete surface damage detection should be referenced in the Introduction. For example: Zhou Q, Qu Z, Ju F. A multi‐scale learning method with dilated convolutional network for concrete surface cracks detection[J]. IET Image Processing, 2022, 16(5): 1389-1402.

6.      The content of Introduction part should be focused on concrete surface damage detection not CNN models.

7.      Fig. 6,7,8 in this paper and fig. 4,5,6 in reference [7] are very similar.

Reviewer 2 Report

Manuscript ID:  applsci-2192849

Title: Multiclass Segmentation of Concrete Surface Damages Using U-Net and DeepLabV3+

Suggestion: Major revision

This manuscript introduces a multiclass encoder-decoder convolutional neural network that detects three types of damage: crack, spalling, and void. In summary, the research is interesting and provides valuable results. Still, the current document has several weaknesses that must be strengthened to obtain a documentary result equal to the publication's value.

(1) At the thematic level, the proposal provides an exciting vision, as the automation of damage detection would be a beneficial resource for engineers. Nevertheless, damage detection is not limited to deep learning and neural networks. This issue is an essential limitation of the aspirations of the proposal, whose limitations should be assumed with more rigour and realism in the development of the argumentation of the manuscript.

(2) The document contains a total of 21 employed references, of which 10 are publications produced in the last five years (48%), 10 in the last 5-10 years (48%) and one undated (4%), implying a total percentage of 96 % recent references. In this way, the total number is insufficient, and their actuality seems low.

(3) The scores are not high enough and convincing. Please clarify.

(4) The abstract is complete and well-structured and explains the document's contents very well. Nonetheless, the part relating to the results could provide numerical indicators obtained in the research.

(5) The first paragraph introducing the research topic may present a much broad and comprehensive view of the problems related to your topic with citations to authority references:

Yunchao Tang; Zhaofeng Huang; Zheng Chen; Mingyou Chen; Hao Zhou; Hexin Zhang; Junbo Sun. Novel visual crack width measurement based on backbone double-scale features for improved detection automation, Engineering Structures 2023, 274: 115158. 

(6) The novelty of the study is not apparent enough. In the introduction section, please highlight the contribution of your work by placing it in context with the work that has been done previously in the same domain. 

(7) Generally, the study of the proposed detection techniques is reasonable, and the explanation of the objectives of the work may be valid. However, the limitations of your work are not rigorously assumed and justified.

(8) Vision technology applications in various engineering fields should also be introduced for a full glance at the scope of related areas. For crack detection, please refer to adversarial networks and improved VGG model related articles. For strain detection, please refer to field strain detected via a novel mark-free vision method related articles.

(9) Chapter 2: Materials and Methods: Why the space dropout deactivates the entire feature map and the principle behind it.

(10) Explains the role of the Adam Optimizer.

(11) From the data in Table 3, when both networks use Spatial Dropout and L1, the improvement to MIoU and F1 does not seem to be much. Please explain why.

(12) I didn't see chapter 4.

(13) Chapter 4: Conclusions: It should mention the scope for further research as well as the implications/application of the study.

(14) I recommend including the limitations regarding the consideration of damage indicated in this review in the limitations assessment. This part of the document can be improved and completed with more rigour.

Reviewer 3 Report

a- It is not common to use exact word of paper's title in keyword.

b- Deep learning is just a tool in your paper. Please revise the abstract as your paper supposed to be focused on concrete surface damage. Additionally, some sentences in the abstract is more related to introduction.

c- Please don't use passive form in the abstract.

d- Some references are missed (e.g. The first paragraph of the Introduction section).

e- In the last paragraph of the paper, the remaining part of the paper is described. For example, the rest of the paper is organized as follow: ???.

f- In dataset sub-section, what do you mean by "1024x1024 pixel resolution"? Is it number of pixels or the spatial resolution of the paper?

g- Why did you use Samsung Galaxy Note 8 or Fujifilm X100? Can other sensors be used?

h- What would happen if you rotate images more than three times (Other angles than 90 degree)?

i- What would happen if we use stereoscopy method (Taking images from two angles and extracting height) for detecting damages?

j- Please add a figure to show the workflow of proposed methodology.

k- F1 or F1 score in Table 3?

l- What is your explanation for sudden fall in Figure 16.(a)?

m- The number of Figures is more than expectation of readers.

n- Conclusion part is like last slide of a presentation. Please rewrite it more descriptive than using numbers.

o- Your paper needs more discussion.

Round 2

Reviewer 1 Report

The authors have revised all the questions.

Reviewer 2 Report

I recommend the publication of this manuscript since the authors have successfully addressed all the comments.